# BioCLIP: A Vision Foundation Model for the Tree of Life

## Abstract

Images of the natural world, collected by a variety of cameras from drones to individual phones, are increasingly abundant sources of biological information. There is an explosion of computational methods and tools, particularly computer vision, for extracting biologically relevant information from images for science and conservation. Yet, currently, most of these are bespoke approaches designed for a specific task and are not easily adaptable or extendable to new questions, contexts, and datasets. We develop the first large-scale multimodal model, BioCLIP, as a foundation for general organismal biology questions on images. We leverage the unique properties of biology as the the application domain for computer vision, namely the abundance and variety of images about plants, animals, and fungi, together with the availability of rich structured biological knowledge. We curate and release TreeOfLife-10M (the largest and most diverse available dataset of biology images), train BioCLIP, rigorously benchmark our approach on diverse fine-grained biology classification tasks, and find that BioCLIP consistently and substantially outperforms existing baselines (by 17% to 20% absolute). Intrinsic evaluation further reveals that BioCLIP has learned a hierarchical representation conforming to the tree of life, shedding light on its strong generalizability.[1]

## 1 Introduction

Digital images and computer vision are quickly becoming pervasively used tools to study the natural world, from evolutionary biology (Borowiec et al., 2022; Lürig et al., 2021) to ecology and biodiversity (Tuia et al., 2022; Beery, 2021; Steenweg et al., 2017). The capability to rapidly convert vast quantities of images from museums (Pearson et al., 2020), camera traps (Beery et al., 2020; 2021; Steenweg et al., 2017; Norouzzadeh et al., 2021; Ahumada et al., 2020), and citizen science platforms (Høye et al., 2021; Nugent, 2018; Sullivan et al., 2014b; Antonelli et al., 2023; McKinley et al., 2017; Sullivan et al., 2014a; Swanson et al., 2015; Parham et al., 2017; Simpson et al., 2014; Van Horn et al., 2015; 2018; Norman et al., 2017) into actionable information (e.g. species classification, individual identification, and trait detection) has accelerated or enabled new advances in tasks such as species delineation (Hansen et al., 2020), understanding mechanisms of adaptation (Hoyal Cuthill et al., 2019; Ezray et al., 2019), abundance and population structure estimation (Høye et al., 2021; Teng et al., 2023; Norman et al., 2017; Araujo et al., 2022), and biodiversity monitoring and conservation (Tuia et al., 2022).

However, applying computer vision to answer any biological question is still a laborious task requiring substantial machine learning expertise and effort—biologists must manually label sufficient data for the specific taxa and task of interest, and find and train a suitable (often bespoke) model for the task. Meanwhile, foundation models (Bommasani et al., 2021) such as CLIP (Radford et al., 2021) and GPT-3 (Brown et al., 2020) have proven their extraordinary capability and value in enabling zero-shot or few-shot learning for a wide range of tasks. A foundation model for biology that can be used for tasks spanning the entire tree of life (Hinchliff et al., 2015; Maddison & Schultz, 2007) will significantly lower the barrier and help democratize AI for biology, empowering scientists and informing conservation efforts.

In this work, we develop the first foundation model for the tree of life. To be broadly useful for real-world biology tasks, this model should meet the following desiderata. First, it should support

---

[1]All the data, code, and models will be publicly released on Github and Hugging Face upon acceptance.

**fine-grained classification**, which necessitates learning a **fine-grained representation** of images of organisms. This is because biology tasks often deal with organisms that are visually similar, e.g., closely related species belonging to the same genus (Pinho et al., 2022) or species that mimic others appearance to gain fitness advantage (Hoyal Cuthill et al., 2019). Second, it should **generalize to the entire tree of life** to the comprehensive extent possible to be a versatile tool for researchers with biological domain expertise. This is needed to ensure it is a not a niche tool but an ally for researchers studying many different clades. Further, it is infeasible to collect training data that covers the millions of known taxa (Hobern et al., 2021; IUCN, 2022), so the model must be generalizable to taxa not present in training data. Finally, due to the high cost of biology data collection and labeling, **strong performance in the low-data regime** (i.e., zero-shot or few-shot) is desired.

To this end, we introduce BIOCLIP, the first foundation model for the tree of life, and outline the conceptual framework, design considerations, and contributions below:

1. **TREEOFLIFE-10M: a large-scale, diverse biology image dataset**. We curate and release the largest-to-date dataset of biology images containing over 10 million images covering 527 thousand taxa in the tree of life. In comparison, the current largest available biology image dataset, iNat21 (Van Horn & Mac Aodha, 2021), contains only 2.7 million images covering 10 thousand taxa. Every image in TREEOFLIFE-10M is labeled with its taxon label to the finest level possible, as well as higher taxonomic ranks in the tree of life. TREEOFLIFE-10M enables training BIOCLIP as well as future biology foundation models.

2. **BIOCLIP: the first foundation model for the tree of life**. With a large-scale labeled dataset like TREEOFLIFE-10M, an intuitive and standard strategy (as adopted by other vision foundation models like ResNet50 (He et al., 2016) and Swin Transformer (Liu et al., 2021)) is to use a supervised classification training objective and learn to predict the taxon label from the input image. However, this strategy fails to recognize and leverage the rich structure of taxon labels—taxa do not exist in isolation but are interconnected in a comprehensive taxonomy. Consequently, a foundation model trained with supervised classification may not generalize well to taxa not covered by the training data, nor would it support zero-shot classification of unseen taxa.

Instead, we propose a novel strategy *combining CLIP-style multimodal contrastive learning with the rich biological taxonomy* for BIOCLIP: we "flatten" the taxonomy from Kingdom to the distal-most taxon rank into a string called *taxonomic name*, and use the CLIP contrastive learning objective to learn to match images with their corresponding taxonomic names. Intuitively, this helps the model generalize to unseen taxa—even if the model has not seen a species, it has likely learned a reasonable representation for that species' genus or family. BIOCLIP also supports zero-shot classification with the taxonomic name of unseen taxa. We further propose, and demonstrate the effectiveness of, a *mixed text type* strategy; by mixing different text types (e.g., taxonomic vs. scientific vs. common names) during training, we retain the generalization from taxonomic names while being more flexibility at test time. For example, one can still use BIOCLIP even when only common names are available.

3. **Comprehensive benchmarking**. We comprehensively evaluate BIOCLIP on 10 fine-grained image classification datasets covering animals, plants, and fungi, including a newly curated dataset focusing on rare species unseen during training. Under zero-shot and few-shot settings, we show that BIOCLIP achieves strong performance and substantially outperforms both CLIP (Radford et al., 2021) and OpenCLIP (Ilharco et al., 2021), leading to an average absolute improvement of **20%** (zero-shot) and **17%** (few-shot). Intrinsic analysis further reveals that BIOCLIP has learned a more fine-grained hierarchical representation over the tree of life, illustrating its superior generalization.

## 2 DATA

Recent work has shown that data quality and diversity is critical when training CLIP models (Fang et al., 2022; Nguyen et al., 2022; Gadre et al., 2023). We curate TREEOFLIFE-10M, the largest and most diverse public dataset for computer vision models in biology. We also compile 10 biologically-relevant fine-grained classification datasets for evaluation.

Table 1: Training data sources used in TREEOFLIFE-10M.

| Dataset | Description | Images | Unique Classes |
|---|---|---|---|
| iNat21 | Citizen scientist labeled image dataset from iNaturalist for fine-grained classification. | 2.7M | 10,000 |
| BIOSCAN-1M | Expert labeled image dataset of insects for classification. | 1.1M | 10,635 |
| EOL | Citizen scientist and expert labeled images and labels aggregated by and downloaded from Encyclopedia of Life. | 6.6M | 518,118 |
| **TREEOFLIFE-10M** | **Largest-to-date dataset of labeled biology images.** | **10.4M** | **527,316** |

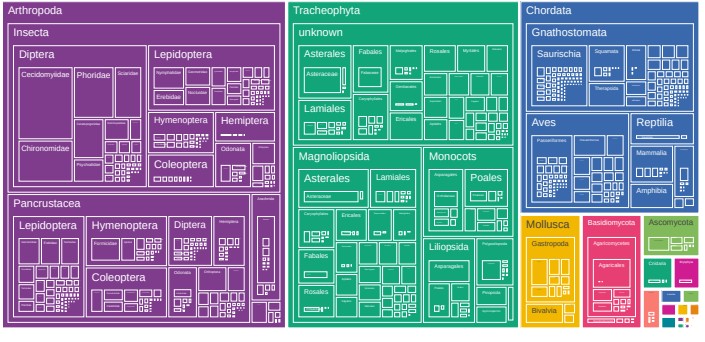
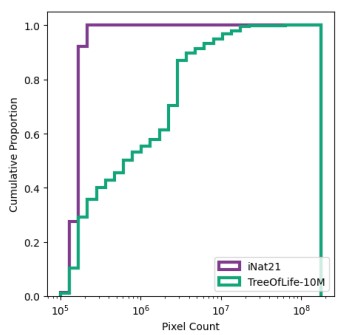

(a) Treemap of the different phyla in TREEOFLIFE-10M. Different colors are different phyla; nested boxes represent classes, orders, etc.

(b) Image size (pixel count, calculated by width×height) cumulative distributions.

Figure 1: 1a shows the overall distribution of images among the 49 phyla present in TREEOFLIFE-10M. 1b shows that TREEOFLIFE-10M has more diversely sized images than iNat21.

## 2.1 TREEOFLIFE-10M

The largest existing biology image dataset is iNaturalist 2021 (Van Horn & Mac Aodha, 2021), which contains 2.7M images of 10K species. Despite this breadth, 10K species is limited for biology. The International Union for Conservation of Nature (IUCN) reported over 2M total described species in 2022, with over 10K bird species and over 10K reptile species alone (IUCN, 2022). iNat21 is thus relatively limited in diversity as a potential dataset to pre-train a foundation model. Furthermore, data diversity is also critical for training high quality visual representations (Nguyen et al., 2023).

Motivated to find high-quality biology image data with a focus on diversity, we turn to the Encyclopedia of Life project (EOL; eol.org). EOL collaborates with a variety of institutions to gather and label millions of images, which are then available for download via an API. These labels connect them within the taxonomic hierarchy to related species. We downloaded 6.6M images from EOL, expanding our dataset by an additional 175K species and 84K genera.

Because insects are such a diverse class (IUCN listed 1M+ unique described insect species in their 2022 report), we also incorporate BIOSCAN-1M (Gharaee et al., 2023), a recent dataset of 1M lab images of insects, covering 494 different families with 3,445 genera and 8,356 species represented. Though BIOSCAN-1M is a rich source of information about insects, it has limitations in its granularity. The vast majority of the datatset is labeled only to the family level (only 22.5% and 7.5% of the data have genus or species indicated, respectively), and 13.6% of the data is even missing this designation. The major contribution of this dataset lies in the rich biological information provided to the level of order, for which all entries have been identified into 19 orders (order is a higher-level taxon represented by the third level of the treemaps in Fig. 1a). Nonetheless, this still introduces a significant amount of diversity to the dataset generating more than 10K unique classes for analysis.

**Aggregation** The final TREEOFLIFE-10M dataset integrates iNat21 (training split), our curated EOL dataset, and BIOSCAN-1M by aggregating the images and unifying the labels. The label integration procedure can be found in Appendix A.

Table 2: Datasets used for evaluation. All tasks are classification evaluated with Top-1 accuracy. We grouped these tasks into broad categories: Animals, Plants & Fungi, and Biologically-Motivated, which are tasks directly related to current biology challenges in computer vision.

| | Name | Description | Examples | Classes | Labels |
|---|---|---|---|---|---|
| Animals | Birds 525 | Scraped dataset of bird images from web search. (Piosenka, 2023) | 2,625 | 525 | Taxonomic |
| | Plankton | Expert-labeled genus and species level in situ images of plankton (Heidi M. Sosik, 2015). | 4,080 | 102 | Mixed |
| | Insects | Expert and volunteer-labeled in-the-wild citizen science images of insects (Serret et al., 2019). | 4,680 | 117 | Scientific |
| | Insects 2 | Mixed common and scientific name classification for insect pests (Wu et al., 2019). | 4,080 | 102 | Mixed |
| Plants & Fungi | PlantNet | Citizen science species-labeled plant images, some drawings (Garcin et al., 2021). | 1,000 | 25 | Scientific |
| | Fungi | Expert-labeled images of Danish fungi (Picek et al., 2021). | 1,000 | 25 | Scientific |
| | Plant Village | Images of leaves on paper, classes are common name healthy or the disease (G. & J., 2019). | 1,520 | 38 | Common |
| | Medicinal Leaf | Species classification of leaves from mature, healthy medicinal plants (S & J, 2020). | 1,040 | 26 | Scientific |
| | PlantDoc | 17 diseases for 13 plant species (Singh et al., 2020). | 1,080 | 27 | Common |
| | Rare Species | Subset of species in the IUCN Red List categories: Near Threatened through Extinct in the Wild (iucnredlist.org). | 12,000 | 400 | Taxonomic |

**Statistics** Table 1 presents dataset statistics. Because different datasets are annotated at different taxonomic levels, we report the number of "unique classes", which are the unique taxonomic names used for training. Fig. 1 shows the distribution of images by phyla and the respective lower-rank taxa (order through family) and the image size distribution of TREEOFLIFE-10M compared to iNat21.

## 2.2 BENCHMARK

We curate a diverse set of 10 biologically-relevant classification tasks to comprehensively benchmark BIOCLIP's potential as a foundation model for the tree of life. Table 2 provides an overview of the datasets; they comprise a variety of label types from full taxonomic names to only scientific or common name, or a mix of the latter.

**Rare Species & Unseen Generalization** Classifying "rare" species is an important and challenging computer vision application in biology, particularly in the context of conservation efforts around the world (Tuia et al., 2022). To the best of our knowledge, there is no publicly available rare species classification dataset; we aim to fill this gap to better leverage computer vision models for biology. To do so, we collect all the species on the IUCN Red List (iucnredlist.org) classification[2] of Near Threatened, Vulnerable, Endangered, Critically Endangered, and Extinct in the Wild. There are approximately 25,000 species that fall into these categories, though image availability is not consistent across species. We select 400 species from the list under the condition that we have at least 30 images per species in our EOL dataset. These species are then completely removed from TREEOFLIFE-10M, creating an *unseen* rare species test set. This dataset demonstrates both 1) BIOCLIP's out-of-distribution generalization on unseen taxa and 2) its potential applications.

**Meta-Album** We extend our benchmark to include biologically-relevant datasets from Meta-Album (Ullah et al., 2022). Meta-Album is a collection of datasets developed for meta-learning, encompassing various subjects, from small animals to plant diseases. Specifically, we use the Plankton, Insects, Insects 2, PlantNet, Fungi, Plant Village, Medicinal Leaf, and PlantDoc datasets. While

---

[2]IUCN has classified 150,388 species and generally updates their list twice per year (IUCN Update Schedule). The classifications used for this dataset are current as of July 13, 2023.

Table 3: Text types considered in the training of BIOCLIP.

| Text Type | Example |
|---|---|
| Common | black-billed magpie |
| Scientific | *Pica hudsonia* |
| Taxonomic | *Animalia Chordata Aves Passeriformes Corvidae Pica hudsonia* |
| Scientific + Common | *Pica hudsonia* with common name black-billed magpie |
| Taxonomic + Common | *Animalia Chordata Aves Passeriformes Corvidae Pica hudsonia* with common name black-billed magpie |

Meta-Album is primarily intended for meta-learning applications, we realize that it can also serve as a testbed for biology foundation models, thanks to its comprehensive coverage.

**Birds 525** Birds 525 (Piosenka, 2023) is a natural choice for bird classification. It is continuously updated and commonly used so that a public record of inaccuracies is maintained for reliability.

### 2.3 DATA DOCUMENTATION AND RELEASE

We will release our curated training data (TREEOFLIFE-10M) and benchmark datasets on HuggingFace (with DOIs) under a public domain license, to the extent primary source licenses allow. For each dataset, this will include a CSV with image metadata with links to the primary source.

## 3 MODELING

BIOCLIP is initialized from the public CLIP checkpoint and pre-trained on TREEOFLIFE-10M with CLIP's multimodal contrastive learning objective.

### 3.1 WHY CLIP-STYLE MULTIMODAL CONTRASTIVE LEARNING?

Compared with general domain computer vision tasks, one of the most salient differences for the biology domain is its rich label space. Not only are the taxon labels large in quantity (there are 2M+ recorded species as of 2022 (IUCN, 2022)), but they are also connected with each other in a hierarchical taxonomy. This poses a great challenge for training a good foundation model that can achieve satisfactory coverage and generalization. On the other hand, the intricate structure in the label space, accumulated through centuries of biology research, provides very rich signals for learning better generalization. Intuitively, if the label space's structure is successfully baked into a foundation model, even if the model has not seen a certain species, it likely will have learned a good representation for that species' corresponding genus or family. Such a hierarchical representation serves as a strong prior to enable few-shot or even zero-shot learning of new taxa.

Most vision foundation models, such as ResNet (He et al., 2016) and Swin Transformer (Liu et al., 2021), adopt a supervised classification objective and directly learn the mapping from input images to class labels. As a result, each class label is treated as a distinct symbol and their relationships are neglected. A key realization of our work is that the multimodal contrastive learning objective used in CLIP can be repurposed for leveraging the hierarchical structure of the label space. Note that this is not an obvious choice; after all, TREEOFLIFE-10M is largely labeled with class labels and not with freeform text like image captions.

CLIP trains two uni-modal embedding models, a vision encoder and a text encoder, to (1) maximize feature similarity between *positive* (image, text) pairs and (2) minimize feature similarity between *negative* (image, text) pairs, where positive pairs are (image, text) pairs from the training data and negative pairs are all other possible (image, text) pairings in a batch. After training, CLIP's encoder models can embed individual instances of their respective modalities into a shared feature space. Next, we discuss how we format the text input to CLIP to incorporate the taxonomic structure.

### 3.2 TEXT TYPES

An advantage of CLIP is the text encoder accepts free-form text. In biology, unlike other classification tasks, class names are diversely formatted. We primarily consider the following:

**Taxonomic name.** A standard seven-level biology taxonomy from higher to lower level is kingdom, phylum, class, order, family, genus and species. For each species, we "flatten" the taxonomy by concatenating all the labels from root to leaf into a single string, which we call the *taxonomic name*.

**Scientific name.** Scientific names are composed of genus and species (e.g., *Pica hudsonia*) and are used as a species' unique identifier.

**Common name.** Taxonomy categories are usually Latin, which is not often seen in generalist image-text pre-training datasets. Instead, the common name, such as "black-billed magpie," is more widespread. Note that common names may not have 1-to-1 mapping to taxa. For a species, there could exist multiple common names. The same common name may also refer to multiple species.

For certain use cases of BIOCLIP, it is possible that only one type of label, e.g., scientific names, is available. To improve the flexibility at inference time, we propose a *mixed text type* training strategy: at each training step, we pair each input image with a text randomly sampled from all of its available text types (shown in Table 3). We empirically show that this simple strategy retains the generalization benefits of taxonomic names while providing more flexibility in using other names at inference time (§4.5). To get the final text input to CLIP, we put the names into the standard CLIP template, e.g., "a photo of *Pica hudsonia*".

## 4 EXPERIMENTS

We hope to understand (1) how dataset diversity and scale affects BIOCLIP's performance and (2) how labels used during BIOCLIP training affect generalization to unseen taxa.

### 4.1 TRAINING AND EVALUATION DETAILS

To train BIOCLIP, we initialize with a CLIP model ViT-B/16 (Radford et al., 2021) that has a vision transformer (Dosovitskiy et al., 2020) image encoder and a 77-token causal autoregressive transformer text encoder. We continue pre-training it on TREEOFLIFE-10M for 100 epochs with a cosine learning rate schedule (Loshchilov & Hutter, 2017). We train on 8 NVIDIA A100-80GB GPUs over 2 nodes with a global batch size of 32,768. We also train a baseline model on only the iNat21 dataset and six ablation models on 1M examples randomly sampled from TREEOFLIFE-10M (§4.5), following the same procedure for BIOCLIP except with a smaller global batch size 16,384 on 4 NVIDIA A100 GPUs on 1 node. More hyperparameters are in Appendix B.

For zero-shot learning, we follow the same procedure as CLIP. For few-shot learning, we follow SimpleShot (Wang et al., 2019) to use a nearest-centroid classifier. For $k$-shot learning, we first randomly sample $k$ examples for each class and obtain the image embedding from the visual encoder of the pre-trained models. We then compute the average feature vector of the $k$ embeddings as the centroid for each class. All the examples left in the dataset are used for testing. After applying mean subtraction and L2-normalization to each centroid and test feature vector, we choose the class with the nearest centroid to the test vector as the prediction. We repeat each few-shot experiment 5 times with different random seeds and report the mean accuracy in Table 4. Results with standard deviations are reported in Appendix C.

We compare BIOCLIP with the original OpenAI CLIP (Radford et al., 2021) as well as OpenCLIP (Ilharco et al., 2021) trained on LAION-400M (Schuhmann et al., 2021). Intuitively, common names of organisms are most pervasive in the training data of CLIP and OpenCLIP and these models work best with common names. This is also confirmed in our preliminary tests. Therefore, we use common names as class labels for CLIP and OpenCLIP by default, unless that is not available for a dataset. BIOCLIP is able to leverage taxonomic names, so we use taxonomic + common names by default. However, as noted in Table 2, the test datasets come in a variety of labels. Whenever the preferred label type is not available, we just use the labels comes with each dataset.

### 4.2 ZERO-SHOT CLASSIFICATION

Table 4 shows that BIOCLIP substantially outperforms both baseline CLIP models for zero-shot classification. Domain-specific training on iNat21 leads to strong improvements on Plant & Fungi tasks: iNat21 has 4.6K plant and fungi species with 1.2M images. TREEOFLIFE-10M leads to further improvements across the board: BIOSCAN-1M adds 1.1M insect images to iNat21's 663K

Table 4: Zero-, one- and five-shot classification top-1 accuracy for different CLIP models. **Bold** indicates best accuracy. All models use the same architecture: ViT-B/16 vision encoders, 77-token text encoder. "iNat21 Only" follows the same procedure as BIOCLIP but uses iNat21 instead of TREEOFLIFE-10M. $\Delta$ denotes the difference in mean accuracy with CLIP.

| Model | Animals | | | | Plants & Fungi | | | | | | |
| | Birds 525 | Plankton | Insects | Insects 2 | PlantNet | Fungi | PlantVillage | Med. Leaf | PlantDoc | Rare Species | Mean ($\Delta$) |
|---|---|---|---|---|---|---|---|---|---|---|---|
| Random Guessing | 0.2 | 1.2 | 1.0 | 1.0 | 4.0 | 4.0 | 2.6 | 4.0 | 3.7 | 0.3 | 2.2 |
| *Zero-Shot Classification* | | | | | | | | | | | |
| CLIP | 49.9 | 3.2 | 9.1 | 9.8 | 58.5 | 10.2 | 5.4 | 15.9 | 26.1 | 26.6 | 21.4 | – |
| OpenCLIP | 54.7 | 2.2 | 6.5 | 9.6 | 50.2 | 5.7 | 8.0 | 12.4 | 25.8 | 31.0 | 20.6 | −0.8 |
| BIOCLIP | **74.7** | **5.4** | **32.7** | **21.2** | **91.0** | **51.8** | **24.0** | **48.1** | **27.5** | **39.2** | **41.5** | +20.1 |
| – iNat21 Only | 55.7 | 2.7 | 29.9 | 12.0 | 89.3 | 42.7 | 16.4 | 22.2 | 18.8 | 19.4 | 30.9 | +9.7 |
| *One-Shot Classification* | | | | | | | | | | | |
| CLIP | 43.7 | 25.1 | 21.6 | 13.7 | 42.1 | 17.2 | 49.7 | 70.1 | 24.8 | 28.4 | 33.6 | – |
| OpenCLIP | 53.7 | 32.3 | 23.2 | 14.3 | 45.1 | 18.4 | 53.6 | 71.2 | 26.8 | 29.3 | 36.7 | +3.1 |
| BIOCLIP | 71.5 | **32.4** | **54.9** | **21.9** | 63.9 | **38.7** | **62.2** | **81.8** | **33.1** | **45.8** | **50.6** | +17.0 |
| – iNat21 Only | **74.6** | 28.9 | 53.8 | 19.4 | **66.8** | 36.3 | 54.3 | 75.9 | 28.2 | 37.2 | 47.5 | +13.9 |
| *Five-Shot Classification* | | | | | | | | | | | |
| CLIP | 73.5 | 41.2 | 39.9 | 24.6 | 65.2 | 27.9 | 71.8 | 89.7 | 35.2 | 46.9 | 51.5 | – |
| OpenCLIP | 81.9 | **52.5** | 42.6 | 25.0 | 68.0 | 30.6 | 77.8 | 91.3 | 42.0 | 48.4 | 56.0 | +4.5 |
| BIOCLIP | **89.9** | 51.1 | **77.2** | **34.4** | 83.6 | **60.6** | **81.8** | **96.0** | **47.6** | **66.4** | **68.8** | +17.3 |
| – iNat21 Only | 89.8 | 47.7 | 73.2 | 32.2 | **84.5** | 55.6 | 76.4 | 93.7 | 40.5 | 56.0 | 64.9 | +13.4 |

insect images, and EOL adds 1.7M plant and fungi images across 89K unique species. In short, we attribute much of BIOCLIP's strong zero-shot performance on this broad and diverse set of tasks to the broad and diverse image and classes present in TREEOFLIFE-10M. With multimodal contrastive learning (§3.1) and the incorporation of taxonomic structure(§3.2), BIOCLIP can capitalize on this data diversity for strong zero-shot generalization.

## 4.3 FEW-SHOT CLASSIFICATION

While labeling data is expensive in specialized domains like biology, biologists naturally label several instances of a given species for resources such as field guides or museum collections. Therefore, we also evaluate BIOCLIP's performance on few-shot classification on the same set of biology datasets with 1 and 5 shots, in Table 4.

We find that BIOCLIP also substantially improves over CLIP baselines in few-shot accuracy. Notably, while Radford et al. (2021) find that CLIP one-shot and two-shot classification is often worse than zero-shot (because few-shot settings cannot use the semantic information in the class name), BIOCLIP has learned useful visual representations that are useful even with only one labeled example: BIOCLIP's mean one-shot accuracy is 9.1% higher than zero-shot accuracy.

## 4.4 INTRINSIC EVALUATION

BIOCLIP demonstrates strong performance in the low-data regime on our extrinsic evaluation, but why? We further conduct an intrinsic evaluation and directly visualize the image feature representations BIOCLIP has learned to shed light on this question (Fig. 2). The visualization is based on iNat21's validation set, which contains 100K images **not seen during training** of BIOCLIP. We reduce feature dimensionality from 512 to two using t-SNE (Van der Maaten & Hinton, 2008), then color the points based on the image's taxonomic label. For each plot, we run t-SNE independently on the subset of examples under the labeled taxonomical rank. Each plot visualizes one rank of the

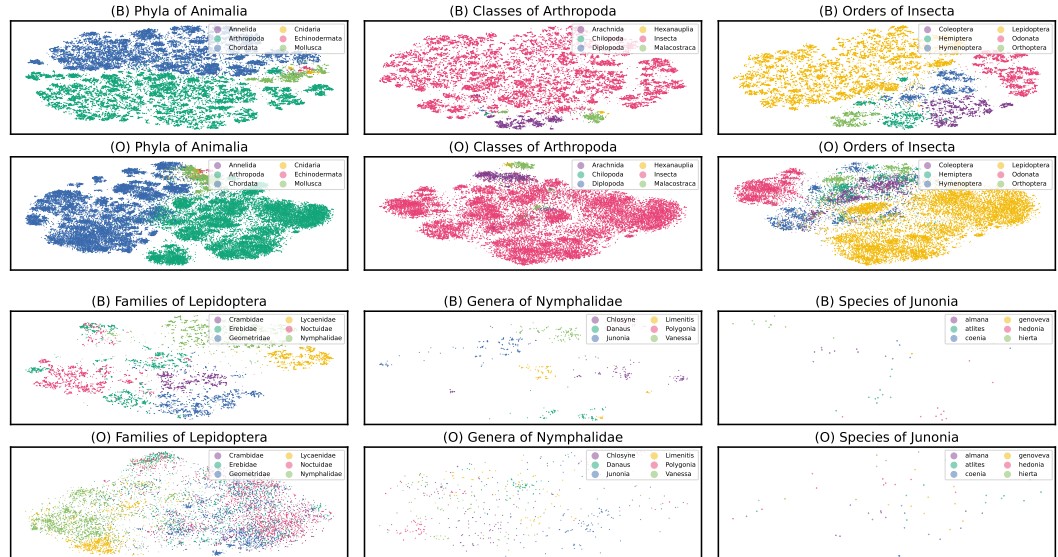

Figure 2: T-SNE visualization of image features, colored by taxonomic labels. CLIP features are visualized in rows two and four with label (O) and BIOCLIP in rows one and three with label (B). At each taxonomic rank, we only plot the six most common classes to avoid visual clutter.

taxonomic hierarchy and the top six categories, based on number of examples, of the following rank, e.g., the top left plot visualizes the top six phyla of the Animalia kingdom.

At higher ranks like kingdom (omitted for space) and phylum, both CLIP and BIOCLIP have good separation, but one can already see that BIOCLIP's representation is more fine-grained and contains a richer clustering structure. At lower ranks, BIOCLIP produces evidently more separable features, while CLIP's features tend to be cluttered and lack a clear clustering structure. This shows that BIO-CLIP has learned a rich feature representation following the hierarchical structure of the taxonomy, which helps explain its strong generalization across the tree of life.

### 4.5 ABLATION ON TEXT TYPES

We conduct an ablation study on the impact of text types used in both training and test by training BIOCLIP on a 10% subset of TREEOFLIFE-10M (10% due to computational constraints). We use our Rare Species benchmark because the test classes have every text type, and all species are excluded from training, making it ideal for testing generalization to unseen taxa. Nguyen et al. (2022) find that the diversity of captions makes stronger vision models and Santurkar et al. (2022) randomly use one of five different captions for each image during training rather than a single fixed caption. Similarly, we use a mixed text type strategy (§3.2). How does that affect model performance?

The zero-shot classification ablation results are in Table 5; there are several salient observations. First, using taxonomic + common names yields the strongest performance, showing the importance of incorporating the taxonomic structure for generalization. Second, when only using a single text type for training, performance degrades substantially when a different text type is used at test time. Using mixed text types for training, while not the strongest, yields consistently strong performance across the board. These results indicate that mixed text type pre-training largely retains the generalization benefits of using taxonomic names while also providing flexibility of different text types for inference, an important property for a foundation model that may be used for diverse downstream tasks. Finally, using 1M examples from TREEOFLIFE-10M outperforms using 2.7M examples from iNat21, further confirming the importance of the added data diversity from TREEOFLIFE-10M.

## 5 RELATED WORK

**Multimodal Foundation Models** CLIP (Radford et al., 2021) trained state-of-the-art vision models from noisy, web-scale (100M+) image-text datasets using a contrastive pre-training objective that

Table 5: Zero-shot accuracy on rare species unseen during training. Green indicates best accuracy and Orange indicates second-best accuracy. Using the taxonomic name over the scientific name always improves accuracy (see 23.2→25.6 and 27.9→28.9). The final rows uses the full iNat21 dataset and TREEOFLIFE-10M for reference. Note that training on 1M images over XX classes (TreeOfLife-1M) outperforms training on 2.7M images over 10K classes (iNat21).

| Dataset | ↓Train/Test→ | Com. | Sci. | Tax. | Sci. + Com. | Tax. + Com. |
|---|---|---|---|---|---|---|
| | Common | 25.7 | 11.0 | 4.5 | 23.7 | 19.3 |
| | Scientific | 12.3 | 23.2 | 8.3 | 20.4 | 11.1 |
| TreeOfLife-1M | Taxonomic | 10.5 | 15.4 | 25.6 | 15.7 | 23.6 |
| | Sci. + Com. | 24.9 | 15.0 | 9.0 | 28.7 | 23.9 |
| | Tax. + Com. | 22.6 | 12.8 | 20.1 | 25.8 | 30.4 |
| | Mixture | 25.2 | 21.0 | 23.4 | 27.9 | 28.9 |
| iNat21 (2.7M) | Mixture | 14.7 | 15.6 | 20.6 | 19.8 | 19.4 |
| TREEOFLIFE-10M | Mixture | 32.9 | 32.7 | 35.6 | 37.8 | 39.2 |

optimized for image retrieval. ALIGN (Jia et al., 2021) and BASIC (Pham et al., 2023) further scaled the number of training examples from 400M to 6.6B, improving vision representation quality.

However, dataset size is not as important as dataset *diversity* (Fang et al., 2022; Nguyen et al., 2022). TREEOFLIFE-10M directly addresses this issue, adding over 500K classes to iNat21's 10K.

**Domain Specific Data** Domain-specific training often outperforms general training (Gu et al., 2021; Chia et al., 2022). This means smaller, domain-specific datasets like TREEOFLIFE-10M are relevant and useful despite the existence of datasets like COCO (Chen et al., 2015), YFCC100M (Thomee et al., 2016), LAION-400M (Schuhmann et al., 2021) or LAION-5B (Schuhmann et al., 2022).

Unfortunately, labeling domain-specific datasets can be prohibitively expensive because annotators must be subject-matter experts. Image-text training is thus particularly potent because annotations are not required; models can learn from noisy image-text pairs. For example, both Ikezogwo et al. (2023) and Lu et al. (2023) gather 1M+ image-text pairs for use in computational pathology, where expert-labeled examples are difficult to gather due to both time and privacy. We gather $10\times$ the images, combining data from multiple sources to further improve image diversity.

Nakkab et al. (2023) use the iNat21 dataset with to train a CLIP model with LiT fine-tuning on the text encoder only. Indeed, they improve top-1 and top-5 accuracy with this approach, but only the iNat21 benchmark is evaluated, so it is unclear whether this performance is maintained across diverse and challenging applications relevant to research such as our Rare Species dataset.

**Domain-Specific Benchmarking** High-quality benchmarks lead to consistent progress. Domain-specific benchmarks are especially challenging because they require expensive, accurately labeled data. Guha et al. (2023) develop a benchmark for legal NLP models by crowdsourcing tasks and Zhang et al. (2023) compile existing benchmarks for biomedical vision-language models. We compile existing benchmarks and develop a biologically-motivated benchmark in our rare species task.

# 6  CONCLUSION

We introduce TREEOFLIFE-10M and BIOCLIP, a large-scale diverse biology image dataset and the first foundational model for the tree of life, respectively. Through extensive benchmarking, we show that BIOCLIP is a strong fine-grained classifier for biology in both zero- and few-shot settings. We corroborate our hypothesis that using the entire taxonomic name leads to stronger generalization than other caption types through ablation studies on unseen species and by visualizing BIOCLIP's representations, finding that images embedded by BIOCLIP better match the hierarchy in the evolutionary taxonomy.

Although we leverage the CLIP objective to efficiently learn visual representations over hundreds of thousands of taxa, BIOCLIP is fundamentally still trained with a classification objective. In future work, we will collect richer textual descriptions of species' appearances such that BIOCLIP can extract fine-grained trait-level representations.

## REPRODUCIBILITY STATEMENT

We will ensure reproducibility of our results by releasing our dataset, data pre-processing code, training code, evaluation code, code to generate all figures (Figs. 1 and 2) and pre-trained model weights. With these resources, anyone with sufficient compute resources can download the original data, then reproduce the pre-processing, training, and evaluation. For those with limited compute, the pre-trained model weights will enable full reproducibility of our evaluation results (Sections 4.2 and 4.3).

## ETHICS STATEMENT

We are not aware of any ethical issues that arise from our work.

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

APPENDICES

We provide details omitted in the main text:

1. Appendix A: Details of training data aggregation
2. Appendix B: Training hyperparameters
3. Appendix C: Standard deviations for few-shot results

## A    TRAINING DATA AGGREGATION

We aggregate images and labels from the iNat21 training data, BIOSCAN-1M's, and data downloaded from EOL. While every image has at least one taxonomic rank labeled, full taxonomic hierarchies and common names are scraped on a best-effort basis from metadata providers, including iNaturalist (iNaturalist Taxonomy DarwinCore Archive), Encyclopedia of Life (eol.org) and Integrated Taxonomic Information System (ITIS) (itis.gov).

We create a lookup between scientific name and taxonomic hierarchy and a lookup between scientific name and common name. We populate these lookups using the following sources in order of descending prioritization, as earlier sources are considered more authoritative. That is, if a duplicate appears in a later source, it is superceded by the higher priority source: BIOSCAN-1M metadata, EOL aggregate datasets, the EOL graph API, information retrieved using EOL page IDs with the pages API, the full list of taxa provided by iNaturalist, the list of vernacular names provided by iNaturalist, and the iNat21 training set class names.

## B    HYPERPARAMETERS

For all trained models, we use a learning rate of 1e-4, 1,000 warm-up steps, and set weight decay to 0.2. The images are resized to $224 \times 224$ pixels.

## C    STANDARD DEVIATION OF MAIN RESULTS

Table C1: Standard deviation of five runs on animals and rare species in Table 4

| Model | Birds 525 | Plankton | Insects | Insects 2 | Rare Species |
|---|---|---|---|---|---|
| *One-Shot Classification* | | | | | |
| CLIP | $43.7 \pm 0.26$ | $25.1 \pm 0.71$ | $21.6 \pm 1.05$ | $13.7 \pm 1.09$ | $28.4 \pm 0.92$ |
| OpenCLIP | $53.7 \pm 0.52$ | $32.3 \pm 0.63$ | $23.2 \pm 1.58$ | $14.3 \pm 0.67$ | $29.3 \pm 0.68$ |
| BIOCLIP | $71.5 \pm 0.72$ | $\mathbf{32.4 \pm 1.53}$ | $\mathbf{54.9 \pm 1.56}$ | $\mathbf{21.9 \pm 0.44}$ | $\mathbf{45.8 \pm 0.85}$ |
| – iNat21 Only | $\mathbf{74.6 \pm 0.32}$ | $28.9 \pm 0.57$ | $53.8 \pm 0.66$ | $19.4 \pm 0.73$ | $37.2 \pm 1.00$ |
| *Five-Shot Classification* | | | | | |
| CLIP | $73.5 \pm 0.37$ | $41.2 \pm 1.01$ | $39.9 \pm 0.86$ | $24.6 \pm 0.90$ | $46.9 \pm 0.21$ |
| OpenCLIP | $81.9 \pm 0.25$ | $\mathbf{52.5 \pm 0.83}$ | $42.6 \pm 0.82$ | $25.0 \pm 0.83$ | $48.4 \pm 0.62$ |
| BIOCLIP | $\mathbf{89.9 \pm 0.14}$ | $51.1 \pm 0.71$ | $\mathbf{77.2 \pm 0.70}$ | $\mathbf{34.4 \pm 0.79}$ | $\mathbf{66.4 \pm 0.32}$ |
| – iNat21 Only | $89.8 \pm 0.19$ | $47.7 \pm 0.85$ | $73.2 \pm 0.75$ | $32.2 \pm 0.71$ | $56.0 \pm 0.16$ |

Table C2: Standard deviation of five runs on plants and fungi in Table 4

| Model | PlantNet | Fungi | PlantVillage | Med. Leaf | PlantDoc |
|---|---|---|---|---|---|
| *One-Shot Classification* | | | | | |
| CLIP | $42.1 \pm 3.40$ | $17.2 \pm 0.78$ | $49.7 \pm 2.53$ | $70.1 \pm 2.83$ | $24.8 \pm 1.61$ |
| OpenCLIP | $45.1 \pm 3.40$ | $18.4 \pm 1.26$ | $53.6 \pm 0.79$ | $71.2 \pm 3.58$ | $26.8 \pm 1.45$ |
| BIOCLIP | $63.9 \pm 5.61$ | $\mathbf{38.7 \pm 4.03}$ | $\mathbf{62.2 \pm 1.68}$ | $\mathbf{81.8 \pm 2.04}$ | $\mathbf{33.1 \pm 1.17}$ |
| – iNat21 Only | $\mathbf{66.8 \pm 0.46}$ | $36.3 \pm 3.10$ | $54.3 \pm 2.25$ | $75.9 \pm 0.97$ | $28.2 \pm 1.64$ |
| *Five-Shot Classification* | | | | | |
| CLIP | $65.2 \pm 1.25$ | $27.9 \pm 2.54$ | $71.8 \pm 1.46$ | $89.7 \pm 1.45$ | $35.2 \pm 1.59$ |
| OpenCLIP | $68.0 \pm 0.86$ | $30.6 \pm 1.26$ | $77.8 \pm 1.28$ | $91.3 \pm 0.85$ | $42.0 \pm 1.32$ |
| BIOCLIP | $83.6 \pm 0.44$ | $\mathbf{60.6 \pm 2.15}$ | $\mathbf{81.8 \pm 0.64}$ | $\mathbf{96.0 \pm 0.92}$ | $\mathbf{47.6 \pm 0.75}$ |
| – iNat21 Only | $\mathbf{84.5 \pm 1.15}$ | $55.6 \pm 1.84$ | $76.4 \pm 0.66$ | $93.7 \pm 0.97$ | $40.5 \pm 1.84$ |

