# OpenReview forum: "BioCLIP: A Vision Foundation Model for the Tree of Life"
_ICLR.cc/2024/Conference — ICLR 2024 Conference Withdrawn Submission_

### Official Review · Reviewer_oEa6 · 2023-10-21

**Soundness:** 2 fair
**Presentation:** 3 good
**Contribution:** 3 good
**Rating:** 6
**Confidence:** 3

**Summary:**

This is a dataset paper that proposes the largest species-centric vision dataset to date, with 10M images or 0.5M species across diverse taxa. They use this dataset to train a CLIP-style model using images and text corresponding to the species name, enriched with higher level taxonomy (genus, family, etc). This model performs substantially better for zero- and few-shot learning in a diverse set of benchmarks than CLIP or a model trained on iNat21.

**Strengths:**

This is an important task, and the community can certainly profit from the dataset, the pre-trained model and the benchmarks.

**Weaknesses:**

The way in which this dataset is built does not really allow to perform zero-shot learning in the more realistic case in which the evaluation is performed over both the seen and the unseen classes (Generalized ZSL). After all, the only shared elements in the side information used are in the high level taxonomy, meaning that, in the ZSL setting, the model can only say: “this image looks like it belongs to this family”, severely limiting its usability in practice. This limitation is not properly addressed in the paper, which may make the reader think that this dataset is actually appropriate for ZSL when it is not.

**Questions:**

I suggest the authors address this limitation by adding GZSL results (I expect them to be very low) and discussing it openly.

---

> ### Author Response · Authors · 2023-11-18
>
> We sincerely thank the reviewer for their time spent reading and engaging with our work.
>
> We think that our benchmark includes both seen and unseen classes during zero-shot.
>
> The PlantVillage and PlantDoc tasks involve classifying images of leaves with both a species and the particular disease (if any) present based on visual symptoms. Because there are no labels during training with disease names, but there are “healthy” labels seen during training, we believe this is an instance of GSZL, **but we didn’t make it clear in our submission.**

---

### Official Review · Reviewer_5Yaa · 2023-11-01

**Soundness:** 2 fair
**Presentation:** 2 fair
**Contribution:** 2 fair
**Rating:** 6
**Confidence:** 5

**Summary:**

This paper provides a large and diverse biological image dataset (named TREEOFLIFE-10M) and trains a CLIP-type model on the proposed dataset (dubbed BIOCLIP). In comparison with existing biological datasets (such as iNat21), the proposed dataset consists of much more diverse categories and images. By training a transformer on this dataset with the image-text contrastive learning from CLIP, this work provides a vision-language model that achieves leading performance in zero- and few-shot image classification tasks.

**Strengths:**

1. The dataset collected in this paper is so far the largest biological image dataset which consists of over 10M images and 0.5M categories. Such a dataset is potentially very useful for training large models that target biology-related vision understanding tasks.
2. This work provides a detailed analysis of the effectiveness of training CLIP-type models on the collected dataset.

**Weaknesses:**

1. The technical contribution is limited. The main contribution lies in the dataset collection and benchmarking of existing methods.
2. The presentation can be further improved. For example, it would be useful to add some examples with annotations from the dataset.

**Questions:**

Even though this work does not provide much technical novelty, it takes a first step to build a large vision model and curate a highly diverse dataset. The analysis of the text types for the training of BIOCLIP is also interesting.

---

> ### Author Response · Authors · 2023-11-18
>
> We thank the reviewer for their time spent reading and engaging with our work.
>
> We will add examples in the supplementary material in the future.

---

### Official Review · Reviewer_4dhf · 2023-11-03

**Soundness:** 3 good
**Presentation:** 3 good
**Contribution:** 2 fair
**Rating:** 3
**Confidence:** 4

**Summary:**

This is a dataset paper. The paper presents a 10M biological image dataset and pre-train a CLIP model for use in evolutionary biology tasks. The model is also benchmarked on a number of datasets and reports good performance.

**Strengths:**

- A largest-to-date dataset of labeled biology images.
- A CLIP-like model is trained on the proposed dataset and indicates a clear advantage on modeled trained on natural images.
- An extensive benmark is provided.
- The paper is well written and easy to follow.

**Weaknesses:**

- Minor novelty. The dataset is a good contribution but not enough for a algorithmic conference like ICLR.
- I am not surprised by the good performance as the model is spefically trained on images of a certain domain.

**Questions:**

I discourage the publication of the paper in ICLR as I think it is more appropriate for biology-orienated journals or workshops. While a substantial number of experiments have been done, I do not see valuable contribution at the technical level.

I would suggest to model specific visual challenges that only specific to the biological domain into model components or variables. Applying off-the-shelf models to a new dataset is not enough for technical conferences like ICLR.

**Details Of Ethics Concerns:**

N.A.

---

> ### Author Response · Authors · 2023-11-18
>
> We sincerely thank the reviewer for their time spent reading and engaging with our work.
>
> With regards to the novelty, we think our novel use of the CLIP objective naturally embeds taxonomic labels into a hierarchical, dense label space, which then teaches the vision model hierarchical representations. This is not a trivial insight, and we have experimented with both supervised classification and hierarchical supervised classification pre-training objectives, and the CLIP objective massively outperforms both. Our current submission lacks these experiments and we agree that without such experiments, our technical novelty is limited.

---

### Official Review · Reviewer_kxWr · 2023-11-09

**Soundness:** 2 fair
**Presentation:** 4 excellent
**Contribution:** 2 fair
**Rating:** 3
**Confidence:** 4

**Summary:**

This paper considers the problem of learning representations for images of plants and animals. In particular, it uses a CLIP-style approach where an image of a species is paired with a text string corresponding to its identity. (This could be the common name, scientific name, string of all taxonomic levels to which it belongs, etc.) The trained models are evaluated on zero-, one-, and five-shot classification.

Generally I quite like the paper, I enjoyed reading it, and I think it has the potential to be a great contribution. However, it has some issues that need to be addressed before it is ready for archival publication. I think this will be a very cool paper when the results are contextualized a little better, both in terms of the framing of the paper and the content of the experiments.

# References (for later)

@inproceedings{taherkhani2019weakly,
  title={A weakly supervised fine label classifier enhanced by coarse supervision},
  author={Taherkhani, Fariborz and Kazemi, Hadi and Dabouei, Ali and Dawson, Jeremy and Nasrabadi, Nasser M},
  booktitle={Proceedings of the IEEE/CVF International Conference on Computer Vision},
  pages={6459--6468},
  year={2019}
}

@article{bilal2017convolutional,
  title={Do convolutional neural networks learn class hierarchy?},
  author={Bilal, Alsallakh and Jourabloo, Amin and Ye, Mao and Liu, Xiaoming and Ren, Liu},
  journal={IEEE transactions on visualization and computer graphics},
  volume={24},
  number={1},
  pages={152--162},
  year={2017},
  publisher={IEEE}
}

@inproceedings{bertinetto2020making,
  title={Making better mistakes: Leveraging class hierarchies with deep networks},
  author={Bertinetto, Luca and Mueller, Romain and Tertikas, Konstantinos and Samangooei, Sina and Lord, Nicholas A},
  booktitle={Proceedings of the IEEE/CVF conference on computer vision and pattern recognition},
  pages={12506--12515},
  year={2020}
}

@article{xiao2020should,
  title={What should not be contrastive in contrastive learning},
  author={Xiao, Tete and Wang, Xiaolong and Efros, Alexei A and Darrell, Trevor},
  journal={arXiv preprint arXiv:2008.05659},
  year={2020}
}

@article{khosla2020supervised,
  title={Supervised contrastive learning},
  author={Khosla, Prannay and Teterwak, Piotr and Wang, Chen and Sarna, Aaron and Tian, Yonglong and Isola, Phillip and Maschinot, Aaron and Liu, Ce and Krishnan, Dilip},
  journal={Advances in neural information processing systems},
  volume={33},
  pages={18661--18673},
  year={2020}
}

@inproceedings{van2021benchmarking,
  title={Benchmarking representation learning for natural world image collections},
  author={Van Horn, Grant and Cole, Elijah and Beery, Sara and Wilber, Kimberly and Belongie, Serge and Mac Aodha, Oisin},
  booktitle={Proceedings of the IEEE/CVF conference on computer vision and pattern recognition},
  pages={12884--12893},
  year={2021}
}

@inproceedings{sariyildiz2021concept,
  title={Concept generalization in visual representation learning},
  author={Sariyildiz, Mert Bulent and Kalantidis, Yannis and Larlus, Diane and Alahari, Karteek},
  booktitle={Proceedings of the IEEE/CVF International Conference on Computer Vision},
  pages={9629--9639},
  year={2021}
}

@inproceedings{cole2022label,
  title={On label granularity and object localization},
  author={Cole, Elijah and Wilber, Kimberly and Van Horn, Grant and Yang, Xuan and Fornoni, Marco and Perona, Pietro and Belongie, Serge and Howard, Andrew and Aodha, Oisin Mac},
  booktitle={European Conference on Computer Vision},
  pages={604--620},
  year={2022},
  organization={Springer}
}

@inproceedings{ng2022animal,
  title={Animal kingdom: A large and diverse dataset for animal behavior understanding},
  author={Ng, Xun Long and Ong, Kian Eng and Zheng, Qichen and Ni, Yun and Yeo, Si Yong and Liu, Jun},
  booktitle={Proceedings of the IEEE/CVF Conference on Computer Vision and Pattern Recognition},
  pages={19023--19034},
  year={2022}
}

@inproceedings{cole2022does,
  title={When does contrastive visual representation learning work?},
  author={Cole, Elijah and Yang, Xuan and Wilber, Kimberly and Mac Aodha, Oisin and Belongie, Serge},
  booktitle={Proceedings of the IEEE/CVF Conference on Computer Vision and Pattern Recognition},
  pages={14755--14764},
  year={2022}
}

**Strengths:**

* The paper is generally well-written, clear, and free of typos.
* The problem addressed by this paper is framed and introduced well.
* The paper has committed to release models and code.
* The idea of using the full taxonomic string for CLIP is interesting and as far as I know novel. The ablation in Table 5 is also a nice idea.
* The experiments in the paper are generally carefully done, and the evaluation protocols seem solid.

**Weaknesses:**

# [W1] Does the paper live up to its very strong claims?
The paper makes some very strong claims: that it introduces "the first foundation model for the tree of life", that it introduces "the largest-to-date dataset of biology images". I'm not sure either of these claims hold up.

Regarding the first claim, it's not clear to me why this should be regarded as "the first foundation model for the tree of life". See the models (of many sizes) listed on [Papers with Code](https://paperswithcode.com/dataset/inaturalist) - why shouldn't any of them count? What is the justification for the (unstated) cutoff for dataset/model size that makes the proposed model "the first"?

Regarding the second claim, it doesn't seem true. Consider iNaturalist, who make their data publicly available. As of November 9, they have ~102M "research grade" observations. I think this counts as a "dataset of biology images" - anyone with the storage space can download them all and start training on them today.

Separately, I'm not sure it's appropriate to combine three existing datasets and claim it as a novel contribution of the paper. What data curation work was done beyond downloading the three datasets and coming up with a mapping between scientific to common names?

# [W2] Why are key parts of the related literature not discussed?

One of the central pieces of the paper is use of taxonomic information in text part of the CLIP training. There is a large literature on the use of taxonomic information in computer vision, and it is not acknowledged in this work. See e.g. [bilal2017convolutional, sariyildiz2021concept, taherkhani2019weakly] for computer vision in general and e.g. [bertinetto2020making, cole2022label] for specific results on natural world images. This body of work bears directly on claims like those related to Figure 2 and Table 5, which relate to the use of hierarchical information to train the model. There is also work on the topic of fine-grained representations for plants and animals in the context of contrastive learning - see e.g. [xiao2020should, cole2022does].

# [W3] Are the experiments in the paper broad enough to establish "foundational" status?

The paper claims to introduce a "foundation model", which the authors state should be "easily adaptable or extendable to new questions, contexts, and datasets." However, zero-shot and few-shot species classification are the only tasks considered. It has not been shown that this model is a useful starting point for diverse downstream tasks in biology (e.g. traditional species classification, segmentation, detection). There is also no evaluation of fine-tuning, which is arguably one of the most important use cases of a "foundation model". For instance, [van2021benchmarking] considers traditional species classification and various biologically inspired "real world" tasks (the NeWT tasks). For another example, [ng2022animal] learns from video and studies grounding, action recognition, and pose estimation. It is these kinds of diverse tasks that constitute evidence that models are "easily adaptable or extendable to new questions, contexts, and datasets".

# [W4] How were common ecology image dataset construction issues handled?
* Taxonomies change often and are generally inconsistent between organizations. This could lead to classes that are duplicated or merged. There is a discussion of the mapping between common and scientific names in the appendix, but no acknowledgement of the fact that the same species can have a different taxonomic path (or even be split or merged with another species) depending on the reference taxonomy being used.
* As far as I can tell, there was no attempt to de-duplicate images, which is important because many organizations source data from other organizations. There may be duplicates or near-duplicates (as in e.g. iNaturalist when people take multiple photos of the same individual) leading to train-test contamination.

# [W5] Why are only CLIP models compared, and is this enough to convince us these models are generally powerful?
The paper only compares variants of CLIP. This means that there is no evidence that the proposed model performs better than e.g. a supervised classifier trained on the same datasets. (After that, an even more fair comparison would be to do supervised training using the full hierarchy of labels - see the hierarchical classification methods discussed above.) In between CLIP and traditional image classifiers, we also have methods like SupCon [khosla2020supervised] which could be used in this setting. While these non-CLIP methods cannot be evaluated for zero-shot, they certainly can for 1- and 5-shot. In addition, all of these methods (CLIP and non-CLIP) can be evaluated on traditional species classification as in the iNaturalist datasets - these results are also missing. It is therefore generally unclear how the proposed model compares to popular and simpler alternatives.

Furthermore, no detail is provided about hyperparameter tuning. This makes it very challenging to interpret comparisons between methods. For instance, maybe the results would look very different with more training epochs or different learning rates. This is especially problematic when the dataset size changes.

**Questions:**

See the weaknesses section - each heading is a question.

# Misc. comments
1. The few-shot performance numbers are generally low in absolute terms. Presumably there is some performance level below which a method is not useful to biologists ("A better than B" -/-> "A is good enough to be useful"). It might be nice to provide the reader with some context along these lines.
2. Table 5 shows that using only the common name is about as good as using the full taxonomic information. Is there any insight into why this is? This seems to invite deeper reflection about what is actually boosting performance when they are combined. What would happen if we permuted the order of the taxon levels before generating all of the strings? What if we permuted them every time you generated one string?
3. Typo: "XX classes" in Table 5 caption.
4. Are there data license issues that should be discussed?
5. The related work on "Domain-Specific Benchmarking" seems a little out of place due to how general it is - that space might be better used for more closely related work.

**Details Of Ethics Concerns:**

It's inappropriate for this paper to have a "no issues" ethics statement. Foundation models in any field are fraught with potential misuses, and ecology is no different. What if the model misleads scientists or policymakers? What if the model is used by poachers? Etc.

---

> ### Author Response · Authors · 2023-11-18
>
> We sincerely thank the reviewer for their time spent reading and engaging with our work. We are especially grateful for the extensive discussion around existing work in both hierarchy and biology in computer vision. We want to address several points:
> * **Why don’t existing models count as vision foundation models for biology?** Foundation models like BERT or GPT-3 quickly became the de-facto model for a wide variety of language-related tasks. We aim for BioCLIP to be the analogous de-facto model for biology-related computer vision tasks. While plenty of large vision models are evaluated on the iNaturalist challenges (CLIP, DINOv2, Hiera), we don’t think they are appropriate for broad use across any arbitrary biology tasks (as evidenced by our evaluation suite). However, we agree that our language does not clearly convey this and will adjust in the future.
> * **Why are key parts of the literature not discussed?** Simply a lack of time in our rush for an ICLR submission. We appreciate the extensive references and will incorporate them in our work in the future.
> * **How were common ecology image dataset issues handled?** Through extensive effort on our part, we canonicalized taxonomic labels across multiple hierarchies. We will include more in-depth discussion in the future.
> * **Why were only CLIP models considered?** Again, a lack of time. We trained both regular supervised classification and hierarchical supervised classification baselines and both underperformed compared to the CLIP models. We will include these results in the future.
> * **Hyperparameter tuning.** We will include details about hyperparameter tuning and the entire training codebase after release.